# Transcriptomic and Metabolic Profiling Reveals a Lignin Metabolism Network Involved in Mesocotyl Elongation during Maize Seed Germination

**DOI:** 10.3390/plants11081034

**Published:** 2022-04-11

**Authors:** Xiaoqiang Zhao, Yining Niu, Xiaodong Bai, Taotao Mao

**Affiliations:** State Key Laboratory of Aridland Crop Science, Gansu Agricultural University, Lanzhou 730070, China; bxd15293898130@163.com (X.B.); m15294392789@163.com (T.M.)

**Keywords:** transcriptome, targeted metabolomics, 24-epibrassinolide signaling, brassinazole signaling, maize seed germination, deep-seeding stress, lignin, peroxisome

## Abstract

Lignin is an important factor affecting agricultural traits. The mechanism of lignin metabolism in maize (*Z**ea mays*) mesocotyl elongation was investigated during seed germination. Maize seeds were treated with 24-epibrassinolide (EBR) and brassinazole stimulation under 3 and 20 cm deep-seeding stress. Mesocotyl transcriptome sequencing together with targeted metabolomics analysis and physiological measurements were employed in two contrasting genotypes. Our results revealed differentially expressed genes (DEGs) were significantly enriched in phenylpropanoid biosynthesis, plant hormone signal transduction, flavonoid biosynthesis, and alpha-linolenic acid metabolism. There were 153 DEGs for lignin biosynthesis pathway, 70 DEGs for peroxisome pathway, and 325 differentially expressed transcription factors (TFs) of MYB, NAC, WRKY, and LIM were identified in all comparisons, and highly interconnected network maps were generated among multiple TFs (MYB and WRKY) and DEGs for lignin biosynthesis and peroxisome biogenesis. This caused p-coumaraldehyde, p-coumaryl alcohol, and sinapaldehyde down-accumulation,
however, caffeyl aldehyde and caffeyl alcohol up-accumulation. The sum/ratios of H-, S-, and G-lignin monomers was also altered, which decreased total lignin formation and accumulation, resulting in cell wall rigidity decreasing. As a result, a significant elongation of maize mesocotyl was detected under deep-seeding stress and EBR signaling. These findings provide information on the molecular mechanisms controlling maize seedling emergence under deep-seeding stress and will aid in the breeding of deep-seeding maize cultivars.

## 1. Introduction

Plant photosynthesis produces about 170 billion tons of lignocellulose each year by immobilizing atmospheric CO_2_ to make plant secondary cell walls [1]. Lignin is a phenolic heteropolymer that is common in plants, and its structure and content vary among species, individuals, tissues, cell types, cell layers, and environments, and plays important biological functions [2]. It is also an attractive target for conversion into biomaterials and biofuels due to its abundance and renewability [3].

Lignin is one of the most important secondary metabolites and is produced by the phenylpropanoid pathway in plant cells. Its biosynthesis involves a very complex network (Appendix A), and is divided into a series of reactions, including deamination, hydroxylation, methylation, and reduction. After it is synthesized, lignin monomers in cytoplasm are transported to the apoplast, and further lignin is randomly polymerized with three types of monolignols, i.e., *p*-coumaryl alcohol (H-unit), coniferyl alcohol (G-unit), and sinapyl alcohol (S-unit) by peroxidase (POD) and laccase (LAC) in the secondary cell wall [4,5,6,7]. Recent evidences suggest lignin level determines cell wall rigidity and relaxation and play roles in hydrophobic properties, mineral transport through vascular bundles [8,9], and responses to plant pathogens and environmental stresses [10,11]. It also regulates plant growth and development [12,13]. Fortunately, key genes in lignin biosynthesis have been isolated, and their biological functions have been verified in several plants in recent years. For example, phenylalanine ammonia-lyase (PAL) is the first key enzyme in lignin biosynthesis, the stems cell walls of *BdPAL*-knockdown grass *Brachypodium distachyon* plants by RNA interference (RNAi) reduced 43% and 57% lignin and ferulate content, respectively, and delayed plant development and root growth [14]. 4-coumarate-CoA ligase (4CL) is the last key enzyme in phenylpropane metabolic pathway, and the inhibition of *Os4CL3* gene expression significantly reduced lignin content and plant height in *Oryza sativa* [15]. Trans-cinnamate 4-monooxygenase (C4H) belongs to the cYP73 subfamily of plant cytochrome P450 as a monooxidase, and a decrease in its activity in transgenic *Nicotiana tabacum* led to lignin levels reduction, *C4H* gene expression also changed lignin composition, resulting in a decrease in the S-/G-unit ratio but no abnormal growth [16]. As is well known, organelles and peroxisomes are enriched in various enzymes, including superoxide dismutase (SOD) and catalase (CAT), which can oxidize substrates and produce reactive oxygen species (ROS; H_2_O_2_, −OH, O_2_^•−^) and H_2_O_2_ hydrolysis reaction. H_2_O_2_ induced POD activity, which then oxidizes *p*-coumaryl-/coniferyl-/sinapyl-alcohol on the cell wall to polymerize into lignin monomers [6,17]. *Arabidopsis* dual transgenic lines with *PaSOD* and *PaAPX* (ascorbate peroxidase) enhanced lignin deposition in their vascular bundles with an altered S-/G-unit ratio under salt stress [18]. Hence, peroxisomes may involve in lignin metabolism. Intriguingly, some transcription factors (TFs; MYB, NAC, and WRKY) can also work as regulators of lignin biosynthetic genes [19,20,21,22].

Maize (*Zea mays*) is a primary cereal crop, and is widely cultivated for food, animal feed, and industrial materials. Since the soil moisture of 0~10 cm soil layer accounts for approximately 15% of the necessary water when maize seeds are sowed in (semi)-arid areas, they are constantly exposed to drought stress, from seed germination to the emergence of seedlings [23]. In agricultural practice, the seeds reached the moist soil required for better seedling emergence and absorbed water from the deep soil layer. For example, *Triticum aestivum* coleoptile (COL) [24] and maize mesocotyl (MES) [25,26,27] markedly elongated to push the shoots to the surface, and then finally ensured normal emergence after deep-seeding, which is an effective strategy for improving their drought tolerance. Moreover, multiple phytohormones signal transduction and their interaction networks were involved in maize MES elongation under deep-seeding stress [6,23,28]. When maize MES and COL were cultivated in light and deep soil environments, their lignin deposition also caused a thicker cell wall, and MES and COL elongation was inhibited by the cell’s turgor pressure [6,9,17]. However, little is known concerning the key genes for lignin biosynthesis pathways or the regulating networks in maize MES elongation when it is exposed to deeper soil layers. Hence, it is essential to identify potential genes and TFs to improve maize MES deep-seeding tolerance. We selected deep-seeding tolerant versus intolerant maize genotypes, and we focused our analyses on the variation in lignin biosynthesis pathways of maize MES. We compared the levels of various intermediate metabolites as well as the genes/TFs of peroxisome pathways involved in lignin biosynthesis between two contrasting maize MES. Our results provide information on the molecular pathway of lignin biosynthesis in maize MES and candidate genes/TFs to target the manipulation of lignin content, providing a strong reference for breeding deep-seeding tolerant maize cultivars.

## 2. Results

### 2.1. Morphological and Physiological Responses to DSS, EBR and BRZ Signaling in Maize MES

Maize MES of deep-seeding tolerant N192 and intolerant Ji853 was incubated for 10 days at 3 cm sowing depth (CK), 20 cm sowing depth (DSS), 4.16 × 10^−3^ M 24-epibrassinolide [EBR; an active brassinosteroid (BR)] induction at 20 cm sowing depth (DSS + EBR), and 4.16 × 10^−3^ M EBR and 3.05 × 10^−3^ μM brassinazole (BRZ; an inhibitor of BR biosynthesis) induction at 20 cm sowing depth (DSS + EBR + BRZ) conditions, these conditions led to significant (*p* < 0.05) changes in morphology and physiology (Figure 1A–C).

Unlike CK, DSS induced an average elongation of mesocotyl length (MESL) of 93.61% of the two maize genotypes, and their average mesocotyl weight (MESW), malondialdehyde (MDA) content, H_2_O_2_ content, and SOD activity increased by 85.82%, 165.79%, 80.66%, and 41.88%, respectively; however, their average POD activity, CAT activity, and lignin content decreased by 18.71%, 21.25%, and 19.57%, respectively (Figure 1B,C). After the two genotypes were treated with 4.16 × 10^−3^ M EBR at 20 cm DSS for 10 days, their average MESL, MESW, SOD activity, POD activity, and CAT activity markedly increased by 16.72%, 15.49%, 27.27%, 1.52%, and 25.71%, respectively. Their average H_2_O_2_ level, MDA content, and lignin content, on the other hand, decreased by 47.71%, 43.65%, and 21.29%, respectively (Figure 1B,C). In addition, when the N192 and Ji853 seeds were germinated 10 days under both 3.05 × 10^−3^ μM BRZ and 4.16 × 10^−3^ M with EBR inductions at 20 cm sowing depth, their average MESL (36.36%), MESW (35.10%), SOD activity (15.32%), and CAT activity (11.96%) were significantly lower. Conversely, their average H_2_O_2_ level (77.10%), MDA content (33.14%), POD activity (11.20%), and lignin accumulation (35.91%) significantly increased (Figure 1B,C). 

Intriguingly, correlation analysis showed MESL/MESW had a significantly negative correlation in terms of lignin content and POD activity, and displayed an obviously positive correlation with SOD activity (Figure 1D). Furthermore, lignin content showed a clearly positive correlation with H_2_O_2_ level and POD activity, and had a distinctly negative correlation with SOD activity. In addition, H_2_O_2_ level displayed a visibly positive correlation with MDA content, and a significantly negative correlation with CAT activity (Figure 1D). These findings led to the speculation that EBR/BRZ was the positive/negative regulator of maize MES growth, and that the lignin metabolism level is a critical regulatory factor in terms of elongation under different treatments. The activities of SOD and POD and lignin accumulation directly affect maize MES growth, while the deposition and decomposition of ROS also affects lignin metabolism level and indirectly affects its growth under DSS, exogenous EBR and BRZ stimulation. 

### 2.2. Analysis of Intermediate Metabolites Involved in Lignin Biosynthesis

Samples of MES in N192 and Ji853 under CK, DSS, and DSS + EBR conditions were collected. These were used to identify and quantify the intermediate metabolites involved in lignin biosynthesis. The principal component analysis (PCA) scores for samples were a good fit (R^2^ = 0.720) and exhibited good predictive value (Q^2^ = 0.361) in the PCA scoring diagram (Figure 2A). The significant principal component (PC) 1 contained 55.2% variables, and the significant PC2 contained 17.0% variables. PCA indicated the distribution of samples was roughly the same and that there was no sample out of the scoring chart that contained the Hotelling’s T2 95% confidence interval. Therefore, this was usable in the subsequent analysis. In all, 25 intermediate metabolites are generally involved in the lignin biosynthesis. In total, 13 metabolites were detected in this study (Figure 1D). Then, metabolites with variable importance in projection (VIP) > 1 and fold change ≥1.5 or ≤0.5 as differential metabolites for group discrimination that were the differentially accumulated metabolites (DAMs). Five DAMs were detected in four comparisons of N192 (DSS_VS_CK; VS1), N192 (DSS + EBR_VS_CK; VS2), Ji853 (DSS_VS_CK; VS3), and Ji853 (DSS + EBR_VS_CK; VS4) (Figure 2C). 

A correlation network analysis (Figure 1D) was performed on the intermediate metabolites of lignin synthesis to further reveal the effects of sowing depths and EBR signaling on lignin synthesis in maize MES. Under DSS and DSS + EBR conditions, lignin level was correlated with caffeate, caffeyl aldehyde, cinnamic acid, ferulic acid, sinapinaldehyde, sinapic acid, p-coumaraldehyde, and p-coumaryl alcohol. Moreover, 30 and 20 pairs of metabolites also exhibited significant correlations. These findings showed these intermediate metabolites participated in lignin biosynthesis through their synergistic and inhibitory effects on each other, and lignin accumulation in maize MES was decreased by the inhibition of the S-unit and H-unit at 20 cm DSS and EBR stimulation. 

### 2.3. Overview of Transcriptome Sequencing and Gene Expression Analysis

To explore the patterns of expression in changes of genes related to maize MES elongation after DSS and DSS + EBR treatments, RNA-sequencing (RNA-Seq) was performed on 18 libraries of MES in N192 and Ji853 in the CK, DSS, and DSS + EBR conditions (three biological replicates for each sample), and then compared to differentially expressed genes (DEGs) that respond to DSS and EBR signaling. After removing low-quality reads, an average of 51.43 million clean reads were obtained from each sample, and the Q30 for all samples were ≥93.18%, while the GC content of each sample was 53.97%, and over 89.5% of clean reads were successfully mapped to the *Zea_mays* B73_V4 reference genome (Appendix A). The PCA of the RNA-Seq data for all samples showed the experiment was reliable, and the samples selection was reasonable (Appendix A). The FPKM density profile was a non-standard normal distribution with a regional area size of 1, representing a sum of approximately 1 for probability (Appendix A). 

### 2.4. DEGs of Maize MES Response to DSS and EBR Stimulation

In this study, the number of DEGs ranged from 3798 (VS3; 2251 upregulated and 1547 downregulated) to 13,449 (VS2; 6181 upregulated and 7268 downregulated), representing in total 23,230 (64.9% of the expressed genes) unique DEGs (Appendix A). The investigation of different genotypes within the deep-seeding tolerant maize as well as the intolerant genotype explained why maize MES elongation varies under different treatments. KEGG pathway analysis was used to classify the complex biological functions corresponding to DEGs. The top 20 KEGG enrichment terms were “phenylpropanoid biosynthesis (map00940)”, “plant hormone signal transduction (map04075)”, “flavonoid biosynthesis (map00941)”, and “alpha-Linolenic acid metabolism (map00592)” (Appendix A). These results implied these pathways, in particular lignin biosynthesis from the phenylpropanoid pathway may play a critical role in maize MES elongation under DSS and EBR signaling.

### 2.5. DEGs for Lignin Biosynthesis Pathway

Lignin is an important structural substance in secondary cell walls, and it controls cell wall elasticity, influences cell wall relaxation, and ultimately determines the growth rate of specific tissue [6]. In total, 153 DEGs encoding multiple enzymes catalyzed lignin biosynthesis through the phenylpropanoid biosynthesis pathway (map00940; Figure 2D) and exhibited differences in their expression levels during MES elongation among the four comparisons. About 58–59% of DEGs were significantly downregulated in both types of maize MES. This suggested that to promote MES elongation and enable normal seedling emergence from deep soil layers, maize MES maintains low lignin synthesis under DSS and EBR induction, which is lower in deep-seeding tolerant maize genotypes (Figure 2B).

### 2.6. Peroxisome Biogenesis Pathway Involved in Lignin Biosynthesis

H_2_O_2_ is a stress-induced ROS and plays an important regulatory role in lignin biosynthesis [29]. Peroxisomes is the main organelle that produce H_2_O_2_. We noticed dramatic changes in the expression of 70 DEGs associated with the peroxisome biogenesis pathway during MES elongation between N192 and Ji853 under DSS and DSS + EBR treatments relative to controls (Appendix A). It was forecast that half of the DEGs could inhibit H_2_O_2_ accumulation, clearly influencing lignin metabolism and positively regulating maize MES plasticity under the DSS and DSS + EBR treatments through their negative expression pattern, but only 5.7% DEGs were differentially expressed and demonstrated their complexity of regulation. The others were upregulated and inversely proportional to the low level of lignin accumulation (Appendix A).

### 2.7. Differentially Expressed TFs Regulated Lignin Biosynthesis

There are abundant TFs in plants, and some, such as MYB, NAC, WRKY, and LIM play an important regulatory role in lignin biosynthesis [4] and ROS homeostasis [30]. In this study, a total of 807 and 797 differentially expressed TFs were identified in deep-seeding tolerant N192 and intolerant Ji853 (Figure 3A). This finding suggested these TFs in tolerant maize MES are more susceptible to activation under DSS and DSS + EBR treatments, thereby regulating the expression of downstream genes in response to DSS and EBR stimulation. Multiple MYB (28 to 77), NAC (23 to 44), WRKY (23 to 53), and LIM (0 to 3) exhibited significant regulation in four comparisons (Figure 3C). They may regulate the expression of lignin synthetases and peroxisome biogenesis genes through their complex interactions with each other to control lignin formation in maize MES. Hence, their interaction networks require deep study. In all of them, 42 co-expressed TFs belonged to AP2/ERF (4), B3 (1), bHLH (7), bZIP (3), C2C2 (2), HB (2), HSF (1), RWP-RK (1), Tify (2), Whirly (1), MYB (9), NAC (6), and WRKY (2) were detected (Figure 3B). Thus, greater attention was paid to the study of these co-expressed TFs to reveal their functions in regulating maize MES development in various environments.

### 2.8. Chromosomal Distribution and Interaction Networks of Identified DEGs and Differentially Expressed TFs 

A total of 153 DEGs of the lignin biosynthesis pathway, 70 DEGs of the peroxisome biogenesis pathway, and 325 differentially expressed TFs of MYB, NAC, WRKY, and LIM were obtained in our transcriptome. Next, their positions were determined by mapping their sequences onto the chromosomes of the *Zea may* B73_V4 genome (Appendix A). The DEGs and differentially expressed TFs were extensively and unevenly distributed across 10 chromosomes, and most were located on chromosomes 3, 1, and 8 (Appendix A). Showing the corresponding positions of these chromosomes could be useful for genetic improvement of maize MES. Multiple DEGs were closely linked with each other (Appendix A), indicating that there may be interactions among these DEGs and differentially expressed TFs. For example, there were highly interconnected networks among multiple DEGs/differentially expressed TFs of MYB, WRKY, 4CL, C4H, ferulate-5-hydroxylase (F5H), and long-chain acyl-CoA synthetase (ACSL); a tight correlation among multiple DEGs of cinnamyl-alcohol dehydrogenase (CAD), (S)-2-hydroxy-acid oxidase (HAO), CAT, and SOD; and a close interaction among multiple DEGs of C4H, coniferyl-aldehyde dehydrogenase (REF1), 2,4-dienoyl-CoA reductase (PDCR), and dehydrogenase/reductase SDR family member 4 (DHSR4). In addition, there were interaction between some DEGs/TFs of acyl-CoA oxidase (ACOX), acetyl-CoA acyltransferase 1(ACAA1), peroxin-13/14 (PEX13/PEX14), MYB, and WRKY (Appendix A).

### 2.9. Quantitative Real-Time (qRT-PCR) Validation of DEGs 

To further analyze the differences in DEGs and differentially expressed TFs at the transcriptional level, we verified the reliability of our RNA-Seq data using qRT-PCR and analyzed the relative expression level of 30 selected DEGs, including 10 DEGs in lignin biosynthesis pathway, 8 DEGs in the peroxisome biogenesis pathway, and 12 differentially expressed TFs (including 6 MYB, 4 NAC, and 2 WRKY) (Figure 4A,B).

## 3. Discussion

Maize seedlings have to struggle out from deep soil after seed germination mainly using significant MES elongation, subsequent to beginning photosynthesis, gradually turning from heterotrophism to autotrophism, and conducting light morphogenesis, and the re-elongation of the MES is inhibited [6,27,31]. Several maize varieties with long MES have been widely applied in semi-arid and arid regions in recent years [32]. It has been found that maize seeds significantly elongate some organs to ensure the rapid germination of seeds and the normal emergence of seedlings [27]. Indeed, the cooperative elongation of maize MES and COL is the crucial response that makes the genotype tolerant to deep-seeding in maize [6,23]. In this study, we also observed that maize MES elongation is the major cause of its deep-seeding tolerance (Figure 1B,C), but the mechanism of this response remains unclear.

Previously, the physiological response mechanisms of maize MES growth have mainly been explored in relation to phytohormone profiling, which has shown that maize MES development is largely regulated by multiple phytohormones [6,9,23]. In addition, as is well known, lignin was largely deposited in the secondary cell wall at later stages of cell differentiation [33], and increasing evidences suggested that increased lignin deposition can enhance cell wall rigidity and decrease cell extension, resulting in inhibition of MES elongation ability under DSS and light stimulation [6,17,23,34]. Our results were consistent with those findings: when the MES of N192 and Ji853 were cultured at a 20 cm depth of soil, lignin accumulation decreased by 28.16% and 10.98% (Figure 1C), and MESL/MESW were closely related to lignin content ranging from −0.945 to −0.936 (Figure 1D), which indicated a low lignin formation. Therefore, breeding for low lignin accumulation in maize MES could contribute to the development of deep-seeding tolerant cultivars. Lignin synthetase of POD plays an important role in dehydrogenation and participates in the polymerization of monolignols into cell wall [4,5,6]. Higher POD activity accelerated lignin accumulation in *Oryza sativa* roots, which slowed their growth under copper stress [29]. We further found POD activity of MES of the two maize genotypes (16.32% in N192 and 21.09% in Ji853) significantly decreased under DSS treatment (Figure 1C). This finding demonstrated maize MES under DSS accumulated low amounts of lignin through a weakened polymerization of lignin monolignols, which may be the mechanism whereby DSS inhibits POD activity. 

H_2_O_2_, as a form of ROS with a relatively long half-life, has a homeostasis that is regulated by a multiple antioxidase system under different abiotic stresses, and it is also an important signaling transduction molecule in a variety of physiological functions [35]. When H_2_O_2_ accumulated in maize MES after light stimulation, the oxidation of lignin monomers induced by POD form lignin, which caused the hardening of the cell wall and inhibition of MES elongation [17]. In agreement with previous studies [6,9], we found the H_2_O_2_ content of MES in nontolerant/tolerant Ji853/N192 had a marked 128.90/32.43% increase, and it was also accompanied by an increase in MDA content (210.74/120.85%) at the level of 20 cm DSS (Figure 1C). Even the increased H_2_O_2_ levels also underwent severe programmed cell death (PCD) and accelerated MES elongation upon exposure to a 20 cm sowing depth [6]. Moreover, maintaining a well-developed enzymatic antioxidant defense system can enable the plant to counter the deleterious effects of ROS. We further found SOD activity (52.33% and 31.44% increase) and CAT activity (8.10% and 34.39% decrease) clearly differed between N192 and Ji853 MES under DSS treatment (Figure 1C). Zhao et al. [6] also demonstrated that lignin content was associated with the level of H_2_O_2_ and the activities of SOD, CAT, and ascorbate peroxidase in maize MES in different environments. Our correlation analysis further showed a close correlation among lignin content, H_2_O_2_ accumulation, SOD activity, and CAT activity (Figure 1D). Consequently, lignin formation may be regulated by the H_2_O_2_-mediated monolignols radical-scavenging function of the enzymatic antioxidant defense system, in particular, SOD, CAT, and ascorbate peroxidase in DSS. 

Earlier works have shown that BR and its stereoisomers participate in the elongation of hypocotyl structures in various species. BR application promoted *Arabidopsis* hypocotyl elongation in the dark [36]; it mediated mechanical properties, and the growth of cell wall hypocotyl segments in *Cucurbita maxima* Duch [37]. It caused an up to a fourfold increase in epicotyl length in *Glycine max*. In response to BR, the *BRU1* gene was highly expressed at the apex of the hypocotyl, and made these sites sensitive to BR-induced elongation [38]. In this study, we found MESL and MESW of N192/Ji853 increased by 1.26/1.07 and 1.19/1.12 times under 4.16 × 10^−3^ M EBR (an active BR) induction that were exposed at a 20 cm sowing depth (Figure 1B,C). This is consistent with Zhao et al. [6], who reported that appropriate EBR-induced cell longitudinal growth promotes maize MES elongation in the deep soil layer. 

Some phytohormones including indole-3-acetic acid [39], gibberellin [40], abscisic acid [41], and methyl jasmonic acid [41] as critical regulatory factors were also involved in plant lignin biosynthesis. Even cell wall lignification in cell suspension systems for switchgrass was induced by BR application [42]. We found that the lignin content of N192 and Ji853 MES were further reduced by 25.00% and 17.57% after being induced by 4.16 × 10^−3^ M exogenous EBR at the 20 cm deep soil layer (Figure 1C). On the other hand, compared to the DSS + EBR treatments, after adding 3.05 × 10^−3^ μM BRZ (an inhibitor of BR biosynthesis), the effect of promoting MES growth disappeared in N192/Ji853, and their MEWL and MESW decreased 34.78/37.93% and 39.94/30.26%, respectively. This was regulated by increasing lignin generation (49.91/21.90%), POD activity (21.05/1.35%), H_2_O_2_ accumulation (34.56/119.64%), and MDA content (37.25/29.04%), and reducing SOD activity (13.20/17.44%) and CAT activity (8.31/15.62%) (Figure 1C). These findings intimated that lignin formation could be significantly regulated by EBR and BRZ under DSS and could participate in maize MES elongation.

It is important to identify the molecular mechanisms of maize MES elongation and deep-seeding tolerance under DSS and EBR stimulation to be able to breed deep-seeding tolerant cultivars. Our transcriptome analysis identified 3798 (VS3) to 13,449 (VS2) DEGs (Appendix A) that may benefit future genome annotation efforts in tolerant N192 and intolerant Ji853 MES. 

For the lignin biosynthesis pathway (Figure 2), we detected 8 DEGs encoding PAL differentially upregulated expression in VS2, VS3, and VS4. *Zm00001d017279* (*GRMZM2G170692*, 3.16-fold), *Zm00001d033286* (7.12-fold), *Zm00001d051166* (*GRMZM2G063917*, 2.91-fold), and *Zm00001d053619* (*GRMZM2G153871*, 2.60-fold) were significantly upregulated in tolerant N192 MES under DSS + EBR treatments compared to controls, and were also activated (2.76- to 13.73-fold) in W64A MES with long MES under DSS and DSS + EBR conditions [6], which resulted in enhancing PAL activity [9]. *Zm00001d017274* (*GRMZM2G074604*, *PAL1*) had been cloned from maize, and it had been proven that it was necessary for lignin biosynthesis in *Arabidopsis* in response to temperature and nitrogen stress [43]. The inhibition of PAL and phenylalanine/tyrosine ammonia-lyase (PTAL) could lead to the accumulation of phenylalanine and tyrosine and contribute to enhance lignin deposition in the cell wall followed by a reduction of maize root growth [44]. Yu et al. [7] reported that a *PTAL* gene in *Hordeum vulgare* (*HOVUSG4914000*) was significantly induced in a lodging-resistant group. Consistently, one *Zm00001d051161* (1.07- and 1.97-fold) encoding PTAL was positively expressed in the two maize MES with EBR application under DSS, instead, the expression of *Zm00001d017274* was reduced in N192 under DSS (−1.63-fold) and Ji853 under DSS (−1.25-fold) and DSS + EBR (−1.64-fold) conditions compared to CK control. C4H, a cytochrome P450-dependent hydroxylase, catalyzed the hydroxylation of the aromatic ring of *t*-cinnamic acid in the *para* position, leading to 4-coumaric acid [45]. In *Glycine max*, exogenous C4H, as an allelochemical, also increased the total lignin content, altering the sum and ratios of the H-, G-, and S-lignin monomers, which led to the stiffening of the cell wall and a reduction in root growth [46]. We identified 2 DEGs controlling C4H that showed higher upregulated expression with EBR induction. This intimated that EBR induced the activation of C4H DEGs and positively regulated S-lignin. Transgenic plants with the *4CL* gene had up to 52% less lignin, a 64% higher S/G ratio, and 30% more cellulose [47]. Our results are consistent with those findings. Unlike the 3 cm treatment, eight of the nine DEGs responsible for 4CL displayed varied upregulated expression (1.08- to 4.11-fold) in both genotypes with/without EBR mediation at a 20 cm depth, resulting in dramatic decreases in lignin content. The differential expression of *F5H* in transgenic switchgrass had antagonistic and synergistic effects on the reduction in S-lignin resulting from O-methyltransferase (COMT) suppression [48]. Transgenic *Medicago sativa* expressing a shikimate O-hydroxycinnamoyltransferase (HCT) antisense construct caused an obvious decrease in lignin generation and a clear change in lignin composition, while also exhibiting significant stunting and lower biomass [49]. *SbCCoAOMT* overexpression in *Sorghum bicolor* increased both soluble and cell wall-bound sinapic acid and ferulic acid, but lignin concentration and composition (S/G ratio) remained unaffected [50]. It is interesting to note that, in our study, one DEGs of caffeoyl-CoA O-methyltransferase (CCoAOMT) and two DEGs of F5H, which were mapped in VS2, showed a low level of upregulation, whereas fifteen DEGs encoding HCT displayed a complex expression pattern in all four groups. These findings explained a great deal about why there was reduced lignin accumulation in DSS and EBR stimulation. Almost half of the DEGs of cinnamoyl-CoA reductase (CCR) and CAD were downregulated, which supported results found in an *Arabidopsis thaliana* ccc triple mutant [51] and CAD1-deficient *Populus* [52]. *GRMZM2G035506* (*Zm00001d002042*) encoding POD7 had been confirmed to be located in a PZE-105098349 single nucleotide polymorphism marker to participate in maize MES elongation at a deep sowing depth [53]. More than half of the DEGs that encoding POD exhibit different levels of downregulation in W64A and K12 MES that were cultured in 20 cm stress and EBR stimulation [6]. Consistent with this, we also found more than 77.0% (54 of 70) of DEGs exhibited varied downregulated expression levels among all comparisons. It should be noted that the results for *Zm00001d046186* (*GRMZM2G380247*) and *Zm00001d046184* (*GRMZM2G471357*) were identical between our study and Zhao et al. [6]. Additionally, the varied expression patterns of four REF1 DEGs, sixteen beta-glucosidase (BGLU) DEGs, and nine UDP-glucosyltransferase (UGT) DEGs were further detected in two maize materials under multiple treatments (Figure 2D). The results will further develop and improve knowledge on the lignin biosynthesis pathway and provide new evidence for maize MES elongation in DSS and EBR stimulation.

Several differentially expressed TFs (e.g., MYB, NAC, WRKY, and LIM) are involved in lignin biosynthesis. *EjODO1* had high sequence homology with *AtMYB20*, could trans-active promoters of *EjPAL1*, *Ej4CL*, and *Ej4CL5*, regulating lignin biosynthesis in developing *Eriobotrya japonica* [54]. The overexpression of *AmMYB308* and *AmMYB330* in *Antirrhinum majus* inhibited the expression of some genes encoding 4CL, C4H, and CAD and leads to less than a 17% reduction in lignin content [55]. *R*. *roxburghii* NAC had a significantly negative correlation with 4CL, HCT, 3′-monooxygenase (C3′H), CCoAOMT, CCR, COMT, CAD, and POD genes [56]. *Nicotiana tabacum NtLIM1* had been found to bind specifically to the PAL-box element and activated the expression of a *β-glucuronidase* gene placed under the control of the promoter of the horseradish POD C2 (*prxc2*) gene [20]. Overexpression of *PtrWRKY19* in *Propulus trichocarpa* with an *AtWRKY12* background did not repress the expression of *AtCCR1*, *AtCOMT1*, *AtF5H1*, *At4CL1*, or *AtC4H* relative to wild-type plants [57]. Overall, our transcriptome analysis identified 807 and 797 differentially expressed TFs in N192 and Ji853 MES (Figure 3A). TFs are useful tools for regulating the expression levels of target genes controlling maize MES development. Overall, it is important to prioritize the potential TFs that will have significant impacts on lignin formation and composition. 

Peroxisomes is a very dynamic and metabolically active organelles and is a very important source of ROS, which was associated with oxygen toxicity but also played a central role in the signaling network that regulates essential processes in cells [58]. Several studies showed H_2_O_2_ generation through a peroxisome biogenesis pathway could significantly mediate lignin formation [6]. In our study, 70 DEGs in the peroxisome biogenesis pathway were further identified between N192 and Ji853 under different conditions (Appendix A). The functions of the PEX family were controversial For example, salt and cadmium stresses upregulated *PEX11a*, *PEX11c*, and *PEX11c* in *Arabidopsis* [59], while neither porin *peroxisomal membrane protein 2* (*PXMP2*) nor *PEX11B* was essential for H_2_O_2_ permeation across the peroxisomal membrane [58]. We found four *PEX14*, two *PEX6*, one *PEX10*, and two *PEX13* were downregulated, and only one *PXMP2* was upregulated in response to DSS and EBR stimulation in MES of N192 and Ji853. These findings suggested that a complex response of different *PEX* and *PXMP2* to various stresses was in play. In addition, consistent with Ma et al. [60], we also found three up-regulated expression of *protein Mpv17* (*MPV17*) may directly cleared excess ROS, and *Zm00001d011761* had similar co-expression in three groups except VS2. The principal ROS regulatory enzymatic systems in plant peroxisomes contain CAT and SOD. Interestingly, in our study, the expression levels of all eleven DEGs encoding SOD increased in both materials after DSS and DSS + EBR induction, and two of three DEGs of CAT were weakly downregulated. 

Thus, a highly interconnected network appeared among multiple DEGs for MYB, WRKY, 4CL, C4H, F5H, CAD, REF1, CAT, SOD, HAO, PDCR, ACSL, DHSR4, ACOX, ACAA1, PEX13, and PEX14 (Appendix A). The identified pathway communities represented potential points of crosstalk. These DEGs then controlled *p*-coumaraldehyde down-accumulation in VS1 and VS3, *p*-coumaryl alcohol down-accumulation in VS2 and VS4, and sinapaldehyde down-accumulation in all four groups, and caffeyl aldehyde up-accumulation in all groups, and caffeyl alcohol up-accumulation in VS1, VS2, and VS4 (Figure 2C). As the total lignin content decreases, altering the sum and ratios of the H-, S-, and G-lignin monomers, cell wall rigidity and relaxation was reduced while a significant elongation of maize MES appears in DSS and EBR signaling (Appendix A). 

## 4. Materials and Methods 

### 4.1. Maize Materials and Growth Conditions

Previously identified deep-seeding tolerant maize genotype N192 and intolerant genotype Ji853 seeds were used in this study [9]. Uniform and plump seeds were surface sterilized with 0.5% sodium hypochlorite for 10 min and then rinsed five times with double distilled water. The sterilized seeds were soaked in three solutions [0 M EBR + 0 M BRZ (S0), 4.16 × 10^−3^ M EBR + 0 M BRZ (S1), and 4.16 × 10^−3^ M EBR + 3.05 × 10^−3^ μM BRZ (S2)] for 24 h in darkness. Then according to the proportion of the single solution (i.e., S0, S1, or S2) and dry vermiculite (100 mL:500 g), were mixed well each other and as three culture substrates. Following that, they were put into PVC tubes (17 cm diameter, 50 cm height). Then 30 soaked seeds were sown evenly, after which 3 cm or 20 cm culture substrate was added until reaching the top of the PVC tubes. The seeds were cultivated in a greenhouse (22 ± 0.5 °C, 12 h/d light, 60% moisture) for germination. The treatments included CK, DSS, DSS + EBR, DSS + EBR + BRZ. After 10 days of germination, the corresponding MES was harvested, frozen in liquid nitrogen, and stored at −80 °C, they were then used for physiological measurements, RNA-Seq, ultra-high-performance liquid chromatography-tandem mass spectrometry (UPLC-MS/MS), and genes qRT-PCR analysis.

### 4.2. Growth Parameters and Physiological Measurements of Maize MES

MESL (cm) and MESW (g) were measured using ruler and electronic scales, respectively. For physiological measurements of MES, 0.5 g MES was homogenized in 5 mL of 0.3% thiobarbituric acid (TBA) and 10% (*v*/*v*) trichloroacetic acid. After incubation at 100 °C for 30 min, mixtures were centrifuged at 12,000 rpm for 10 min to measure MDA content [61]; 0.5 g MES was homogenized in 5 mL of ice-cold trichloroacetic acid (0.1%, *w*/*v*) and then centrifuged at 12,000 rpm at 4 °C for 15 min to collect the supernatant for H_2_O_2_ content measurement [6]; 0.5 g MES was homogenized in 5 mL ethanol (95%, *v*/*v*) and then centrifuged at 10,000 rpm at 4 °C for 10 min, and the sediment was further rinsed three times with ethanol-n-hexane solution (1/1, *v*/*v*) and dried. After that dissolved in 0.5 mL bromide acetyl-glacial acetic acid solution (1/3, *v*/*v*) and then bathed in water for 30 min at 70 °C, and further mixed with 0.9 mL NaOH (2 M), 5 mL glacial acetic acid, and 0.1 mL hydroxylamine hydrochloride (7.5 M) to analyze lignin content [6]; 0.5 g MES was homogenized using a precooled mortar and pestle in 1 mL of ice-cold potassium-phosphate buffer (50 mM, pH 7.0) containing potassium chloride (100 mM), ascorbate (1 mM), β-mercaptoethanol (5 mM), and glycerol (10%, *w*/*v*). Homogenates were centrifuged at 12,000 rpm for 10 min, and supernatants were stored at 4 °C for SOD, POD, and CAT activity assay [9]. 

### 4.3. RNA Extraction, Library Construction, and Illumina Sequencing

For transcriptome analysis, total RNA from the MES of N192 and Ji853 under the CK, DSS, DSS + EBR conditions were extracted using commercial kits (TRIZOL reagent, Invitrogen, Carlsbad, CA, USA) according to the manufacturer’s protocol. The total RNA concentration and quality were determined on a Nanodrop spectrophotometer and via 1% agarose gel electrophoresis. Eukaryotic mRNA was enriched using magnetic beads with Oligo (dT), and mRNA was fragmented using an interrupting reagent. mRNA was used to construct a library using an Agilent 2100 Bioanalyzer (Agilent Technologies, Santa Clara, CA, USA), and the resulting ligation products were sequenced using an Illumina NovaSeq PE150 Sequencer at Nanjing Genepioneer Biotechnologies Company, Nanjing, China.

### 4.4. Sequencing Data and DEGs Analysis

Some low-quality reads and reads containing adapters or poly-N were removed from the raw data to generate clean reads, and these were aligned to the *Zea mays* B73_V4 reference genome (ftp://ftp.ensemblgenomes.org/pub/plants/release-6/fasta/zea_mays/dna/, accessed on 12 December 2021) using HISAT2 (http://ccb.jhu.edu/software/hisat2, accessed on 12 December 2021, v. 2.0.1). The aligned reads of each sample were assembled using Cufflinks [62]. To produce non-redundant transcripts, assembled transcripts from three biological replicates were merged, also using Cufflinks. The expression levels of the merged transcripts were quantified and FPKM values were calculated. DEGs were analyzed using an R Package called DESeq2 in Bioconductor (http://www.bioconductor.org/, accessed on 12 December 2021) for each comparison with the parameters of false discovery rate (FDR) < 0.05 and |log2 fold-change| > 1. Then the identified DEGs were subjected to KEGG analysis (http://www.genome.jp/kegg/, accessed on 12 December 2021), GO enrichment analysis (http://bioinfo.cau.edu.cn/agriGO/, accessed on 12 December 2021), COG analysis (hyyps://www.ncbi.nlm.nih.gov/COG/, accessed on 12 December 2021), and Nr annotation (http://www.ncbi.nlm.nih.gov/pubmed, accessed on 12 December 2021), respectively.

### 4.5. Preparation and UPLC-MS/MS Detection of Lignin Biosynthetic Intermediate Metabolites

According to the differences in the mass spectrometry response signals and following Li et al. [63] by UPLC-MS/MS to prepare MES extracts, we detect and quantified lignin biosynthetic intermediate metabolites at Wuhan Metware Biotechnology Co., Ltd., Wuhan, China. The UPLC and MS/MS detection were performed on Shim-pack UFLC SHIMADZU CBM30A (Kyoto, Japan) and Applied Biosystems 4500 QTRAP (Mundelein, IL, USA). The metabolite quantification was accomplished by using the multiple reaction monitoring (MRM) model. Meanwhile the optimization to voltage (DP), cluster collision (CE), and mass spectrum parameters. The optimized mass spectrometry conditions were as follows: curtain gas 30 psi; collision gas 8; ion spray voltage −5500 V, electrospray ionization (ESI) temperature 550 °C; ion source gas 1:55; and gas 2:55. The lignin pathway intermediates were targeted and 12 corresponding metabolites were successfully detected and quantified, while the others were too low to be determined. Then above data matrices with the ion intensity of metabolites were uploaded to the Analyst 1.6.1 software (AB SCIEX, Vaughan, ON, Canada) for statistical analyses. The supervised multivariate method partial least squares-discriminant analysis (PLS-DA) was used to maximize the metabolome differences between different maize MES. The relative importance of each metabolite to the PLS-DA model was checked using the parameter VIP. Metabolites with VIP ≥ 1 and fold change ≥1.5 or ≤0.5 were considered to be differential metabolites for group discrimination.

### 4.6. Gene Expression Analysis Using qRT-PCR

Primer sequences for 30 genes were designed using Primer Premier v5.0 (Appendix A). A cDNA synthesis kit (TaKaRa, Kusatsu Shi, Japan) was used to transcribe RNA into cDNA. qRT-PCR reactions were carried out on a super real premix plus (SYBR Green) (Tiangen, Shanghai, China) using TransStart Tip Green qPCR SuperMix (TRAN, Beijing, China), according to the manufacturer’s protocol. The relative expression levels of the genes were estimated using the (2^−∆∆Ct^) quantification method with maize *Actin1* (*Zm00001d010159*) [6] serving as the endogenous control for normalization.

## 5. Conclusions

The molecular mechanism of maize deep-seeding tolerance is very complex. By analyzing the transcriptomes, targeted metabolomics data, and physiological results of two maize genotypes (N192 and Ji853) MES, some biochemical pathways, in particular phenylpropanoid biosynthesis, and corresponding candidate genes/TFs were closely associated with maize deep-seeding tolerance, such as 153 DEGs controlling lignin biosynthesis, 70 DEGs involving in peroxisome biogenesis, and 325 TFs (MYB, NAC, WRKY, and LIM) were identified in four comparisons. Among them, the highly interconnected regulatory networks among multiple TFs and DEGs were also generated. This synergistically regulated the significant changes of intermediate metabolites accumulation, enzymes activity, and sum/ratios lignin monomers. Consequently, a significant MES elongation in maize under deep-seeding stress and EBR stimulation. This study provides a theoretical basis for further clarification of the complex regulation mechanism of deep-seeding tolerance of maize and the breeding of deep-seeding tolerant maize cultivars.

## Figures and Tables

**Figure 1 plants-11-01034-f001:**
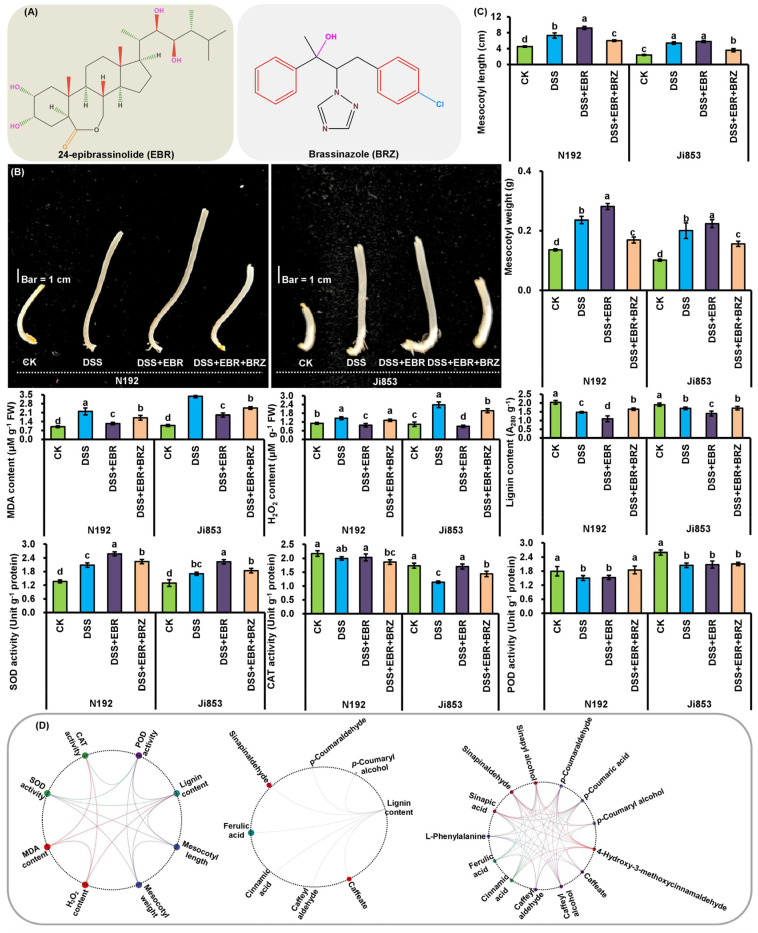
Growth, physiological characteristics, and their relationships of mesocotyls in two contrasting maize genotyes (N192 and Ji853) under 3 cm sowing depth (CK), 20 cm sowing depth (DSS), 4.16 × 10^−3^ M 24-epibrassinolide (EBR) induction at 20 cm sowing depth (DSS + EBR), 4.16 × 10^−3^ M EBR and 3.05 × 10^−3^ μM brassinazole (BRZ) induction at 20 cm sowing depth (DSS + EBR + BRZ). EBR and BRZ structures (**A**). Mesocotyls observation in N192 and Ji853 under four conditions (**B**). Statistics of mesocotyl length (MESL), mesocotyl weight (MESW), malondialdehyde (MDA) content, H_2_O_2_ content, lignin content, SOD activity, CAT activity, and POD activity of mesocotyls in N192 and Ji853 under these conditions. Different lowercase letters with a single maize genotype among four conditions represented significant differences with *p* < 0.05 level (**C**). Interactive ring correlation plot of these traits in both maize genotypes under three conditions (CK, DSS, and DSS + EBR) was performed by a free online Genescloud tool (https://www.genescloud.cn; accessed on 10 December 2021). The connection curve in both traits showed a significant positive or negative correlation in *p* < 0.05 level (**D**).

**Figure 2 plants-11-01034-f002:**
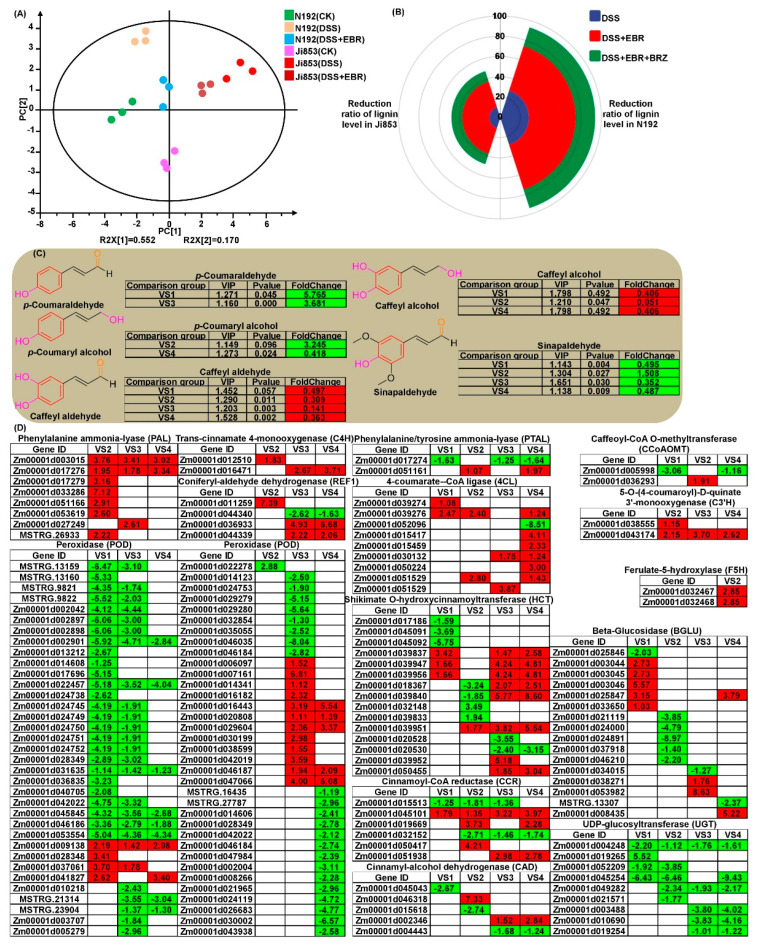
Differentially expressed genes (DEGs) and differentially accumulated metabolites (DAMs) involved in lignin biosynthesis in N192 and Ji853 mesocotyls under CK, DSS, and DSS + EBR conditions. Principle component analysis (PCA) of the lignin metabolites derived from ultra–high–performance liquid chromatography-tandem mass spectrometry (UPLC–MS/MS) (**A**). Polar coordinate graph for reduction ratio of lignin level in N192 and Ji853 mesocotyls under DSS, DSS + EBR, and DSS + EBR + BRZ conditions that compared to CK control (**B**). DAMs were analyzed in all four comparisons [N192 (DSS_VS_CK), VS1], [N192 (DSS + EBR_VS_CK), VS2], [Ji853 (DSS_VS_CK), VS3], and [Ji853 (DSS + EBR_VS_CK), VS4]). The red and green box represented up– and down–accumulated DAMs, and the value in the box was the fold–change of DAMs (**C**). DEGs involved in lignin biosynthesis in four comparisons. The red and green box represented up– and down–regulated DEGs, and the value in the box was the log2 fold-change of DEGs (**D**).

**Figure 3 plants-11-01034-f003:**
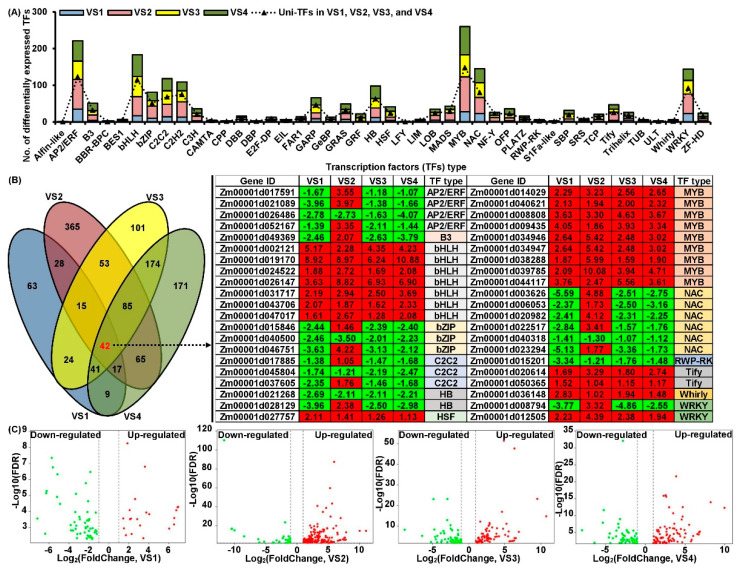
Classification of differentially expressed transcription factors (TFs) in developing N192 and Ji853 mesocotyls under CK, DSS, and DSS + EBR conditions. Statistics of different types of TFs (**A**). Venn diagram of TFs among all comparisons and co–expression patterns analysis of corresponding commonly TFs in all comparisons. The red and green box represented up– and down–regulated differentially expressed TFs, and the value in the box was the log2 fold-change of differentially expressed TFs (**B**). Volcano plot of MYB, NAC, WRKY, and LIM involved in lignin biosynthesis in four comparisons (**C**).

**Figure 4 plants-11-01034-f004:**
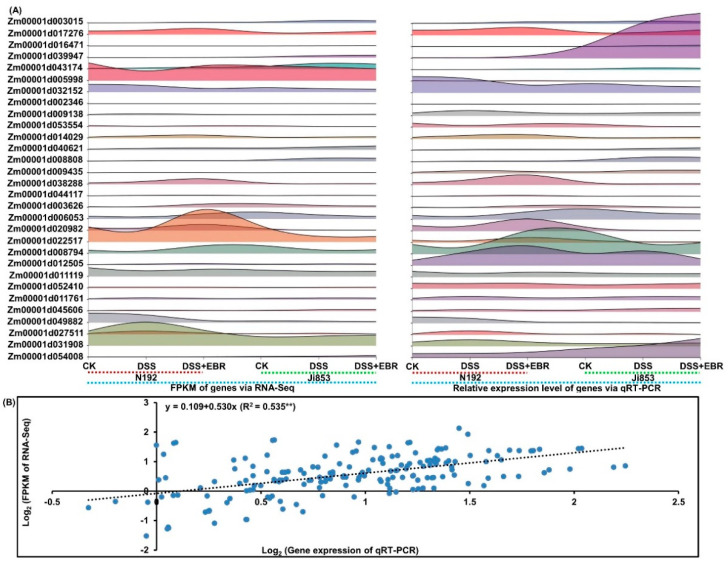
The quantitative real–time (qRT–PCR) expression analyses of 30 selected genes of N192 and Ji853 mesocotyls in CK, DSS, and EBR signaling, respectively. Interactive layered area map for 30 selected DEGs comparisons by qRT–PCR and RNA–-Seq were performed using by a free online Genescloud tool (https://www.genescloud.cn; accessed on 12 December 2021) (**A**). Correlation between qRT-PCR and RNA-Seq data (** *p* < 0.01; ANOVA) (**B**).

## Data Availability

The data is contained within the article and Appendix A.

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
