# Peer review of "Transcriptomic and Metabolic Profiling Reveals a Lignin Metabolism Network Involved in Mesocotyl Elongation during Maize Seed Germination"

_plants, 2022, doi:10.3390/plants11081034_

Round 1

Reviewer 1 Report

Article is intersing, however, some attention is required.

Major issues:

  • Please add aditional informationa about UPLC-MS/MS methodology.
  • How did lignans measure? 
  • Article is too long and some difficult to understanding.

Author Response

Dear Ms. Fiona Yan and Reviewers

Thank you for your letter of – and for the referee’s comments concerning our manuscript, “Transcriptomic and Metabolic Profiling Reveals a Lignin Metabolism Network Involved in Mesocotyl Elongation during Maize Seed Germination (plants-1646395)”. We have carefully studied these comments and have made corresponding corrections to the manuscript, which we describe in detail below. We would like to re-submit the manuscript and that for possible publication on the Special Issue: “Molecular Mechanism of Seed Germination under Different Environment Conditions” of Plants. Thank you very much for your time and consideration.

Your manuscript has now been reviewed by experts in the field. Please find your manuscript with the referee reports at this link: https://susy.mdpi.com/user/manuscripts/resubmit/ 716146146e9aabffde737744ead434c0. Please revise the manuscript according to the referees' comments and upload, the revised file within 10 days. Please use the version of your manuscript found at the above link for your revisions.

Thanks for the positive comments of you and all reviewers for our manuscript. As suggested, we have carefully revised the manuscript. We then have re-submitted the manuscript.

Thank you for your consideration.

(I) Any revisions to the manuscript should be marked up using the “Track Changes” function if you are using MS Word/LaTeX, such that any changes can be easily viewed by the editors and reviewers.

Thanks for the positive comments of you and all reviewers for our manuscript. As suggested, we have carefully revised the manuscript, and all corresponding contents were modified by “Track Changes” function of MS Word. We then have re-submitted the manuscript.

Thank you for your consideration.

(II) Please provide a cover letter to explain, point by point, the details of the revisions to the manuscript and your responses to the referees’ comments.

Thanks for the positive comments of you and all reviewers for our manuscript. As suggested, we have carefully revised the manuscript. In addition, we have prepared a detailed response letter to all reviewers for each point, and then have re-submitted the manuscript.

Thank you for your consideration.

(III) If you found it impossible to address certain comments in the review reports, please include an explanation in your rebuttal.

Thanks for the positive comments of you and all reviewers for our manuscript. As suggested, we have carefully revised the manuscript and have answered all questions of each reviewer point by point. We then have re-submitted the manuscript.

Thank you for your consideration.

(IV) The revised version will be sent to the editors and reviewers.

Thanks for the positive comments of you and all reviewers for our manuscript. As suggested, we have re-submitted the revised version of the manuscript.

 Thank you for your consideration.

Reviewer 1

Major issues:

Please add aditional informationa about UPLC-MS/MS methodology.

Thanks for your positive comments. As suggested, we have re-added the additional information about UPLC-MS-MS methodology, namely “According to the differences in the mass spectrometry response signals and following Li et al. [62] by UPLC-MS/MS to prepare MES extracts, we detect and quantified lignin biosynthetic intermediate metabolites at Wuhan Metware Biotechnology Co., Ltd., Wuhan, China. The UPLC and MS/MS detection were performed on Shim-pack UFLC SHIMADZU CBM30A (Japan) and Applied Biosystems 4500 QTRAP (USA). The metabolite quantification was accomplished by using the multiple reaction monitoring (MRM) model. Meanwhile the optimization to voltage (DP), cluster collision (CE), and mass spectrum parameters. The optimized mass spectrometry conditions were as follows: curtain gas 30 psi; collision gas 8; ion spray voltage −5500 V, electrospray ionization (ESI) temperature 550 °C; ion source gas 1 : 55; and gas 2 : 55. The lignin pathway intermediates were targeted and 12 corresponding metabolites were successfully detected and quantified, while the others were too low to be determined. Then above data matrices with the ion intensity of metabolites were uploaded to the Analyst 1.6.1 software (AB SCIEX, Ontario, Canada) for statistical analyses. The supervised multivariate method partial least squares-discriminant analysis (PLS-DA) was used to maximize the metabolome differences between different maize MES. The relative importance of each metabolite to the PLS-DA model was checked using the parameter VIP. Metabolites with VIP ≥ 1 and fold change ≥ 1.5 or 0.5 were considered to be differential metabolites for group discrimination.” in Lines 508-525 of the manuscript. We then have re-submitted the manuscript.

Thank you for your consideration.

How did lignans measure? 

Thanks for your positive comments. As suggested, we have re-added the lignin measure method, namely “Among them, 0.5 g sample was homogenized in 5 mL ethanol (95%, v/v) and then centrifuged at 10000 rpm at 4 °C for 10 min, and the sediment was further rinsed three times with ethanol-n-hexane solution (1/1, v/v) and dried. After that dissolved in 0.5 mL bromide acetyl-glacial acetic acid solution (1/3, v/v) and then bathed in water for 30 min at 70 °C, and further mixed with 0.9 mL NaOH (2 M), 5 mL glacial acetic acid, and 0.1 mL hydroxylamine hydrochloride (7.5 M) to analyze lignin content [6].” in Lines 475-480 of the manuscript. We then have re-submitted the manuscript.

Thank you for your consideration.

Article is too long and some difficult to understanding.

Thanks for your positive comments. As suggested, we have further revised the manuscript to make it shorter and more readable. We then have re-submitted the manuscript.

Thank you for your consideration.

Reviewer 2 Report

The paper investigated the Lignin Metabolism Network Involved in Mesocotyl Elongation during Maize Seed Germination. The topic is interesting. Although the paper is well organized, the paper is too lengthy and can be as concisely as possible. Besides, too many abbreviations affect the readability of the article.

  1. “Maize MES of N192 and Ji853 was incubated for 10 days at 3 cm sowing depth (CK),…” The authors should indicate which one is deep-seeding tolerant genotypes. The authors should rewrite this paragraph so that it is clear to readers how the authors designed the experiment without looking at the Materials and Methods section.
  2. Add the parameter of HISAT2.
  3. DESeq1 or Seq2?
  4. The abstract was not well prepared and needs to be rewritten.

Author Response

Dear Ms. Fiona Yan and Reviewers

Thank you for your letter of – and for the referee’s comments concerning our manuscript, “Transcriptomic and Metabolic Profiling Reveals a Lignin Metabolism Network Involved in Mesocotyl Elongation during Maize Seed Germination (plants-1646395)”. We have carefully studied these comments and have made corresponding corrections to the manuscript, which we describe in detail below. We would like to re-submit the manuscript and that for possible publication on the Special Issue: “Molecular Mechanism of Seed Germination under Different Environment Conditions” of Plants. Thank you very much for your time and consideration.

Your manuscript has now been reviewed by experts in the field. Please find your manuscript with the referee reports at this link: https://susy.mdpi.com/user/manuscripts/resubmit/ 716146146e9aabffde737744ead434c0. Please revise the manuscript according to the referees' comments and upload, the revised file within 10 days. Please use the version of your manuscript found at the above link for your revisions.

Thanks for the positive comments of you and all reviewers for our manuscript. As suggested, we have carefully revised the manuscript. We then have re-submitted the manuscript.

Thank you for your consideration.

(I) Any revisions to the manuscript should be marked up using the “Track Changes” function if you are using MS Word/LaTeX, such that any changes can be easily viewed by the editors and reviewers.

Thanks for the positive comments of you and all reviewers for our manuscript. As suggested, we have carefully revised the manuscript, and all corresponding contents were modified by “Track Changes” function of MS Word. We then have re-submitted the manuscript.

Thank you for your consideration.

(II) Please provide a cover letter to explain, point by point, the details of the revisions to the manuscript and your responses to the referees’ comments.

Thanks for the positive comments of you and all reviewers for our manuscript. As suggested, we have carefully revised the manuscript. In addition, we have prepared a detailed response letter to all reviewers for each point, and then have re-submitted the manuscript.

Thank you for your consideration.

(III) If you found it impossible to address certain comments in the review reports, please include an explanation in your rebuttal.

Thanks for the positive comments of you and all reviewers for our manuscript. As suggested, we have carefully revised the manuscript and have answered all questions of each reviewer point by point. We then have re-submitted the manuscript.

Thank you for your consideration.

(IV) The revised version will be sent to the editors and reviewers.

Thanks for the positive comments of you and all reviewers for our manuscript. As suggested, we have re-submitted the revised version of the manuscript.

 Thank you for your consideration.

Reviewer 2

The paper investigated the Lignin Metabolism Network Involved in Mesocotyl Elongation during Maize Seed Germination. The topic is interesting. Although the paper is well organized, the paper is too lengthy and can be as concisely as possible. Besides, too many abbreviations affect the readability of the article.

Thanks for your positive comments. As suggested, we have further revised the manuscript to make it shorter and more readable. In addition, we have also deleted some unnecessary abbreviations, such as, AUX (auxin), GA (gibberellin), ABA (abscisic acid), CTK (cytokinin), ET (ethylene), SA (salicylic acid), JA (jasmonic acid), DST (deep-seeding tolerant), IAR4 (IAA alanine resistant), MeJA (methyl jasmonic acid) in the manuscript. We then have re-submitted the manuscript.

Thank you for your consideration.

  1. “Maize MES of N192 and Ji853 was incubated for 10 days at 3 cm sowing depth (CK),…” The authors should indicate which one is deep-seeding tolerant genotypes. The authors should rewrite this paragraph so that it is clear to readers how the authors designed the experiment without looking at the Materials and Methods section.

Thanks for your positive comments. As suggested, we have re-written the corresponding content that “Maize MES of deep-seeding tolerant N192 and intolerant Ji853 was incubated for 10 days at 3 cm sowing depth (CK), 20 cm sowing depth (DSS), 4.16×10-3 M 24-epibrassinolide (EBR) induction at 20 cm sowing depth (DSS+EBR), and 4.16×10-3 M EBR and 3.05×10-3 μM brassinazole (BRZ) induction at 20 cm sowing depth (DSS+EBR+BRZ) conditions, these conditions led to significant (P < 0.05) changes in morphology and physiology (Figure 1A-1C).” in Lines 96-101 of the manuscript. We then have re-submitted the manuscript.

Thank you for your consideration.

  1. Add the parameter of HISAT2.

Thanks for your positive comments. In this study, follow the instructions and parameter settings of HISAT2 software from the HISAT2 graph-based alignment of next generation sequencing reads to a population of genomes website (http://ccb.jhu.edu/software/hisat2, v. 2.0.1) to use the HISAT2 software. Therefore, as suggested, the corresponding contents were revised that “Some low-quality reads and reads containing adapters or poly-N were removed from the raw data to generate clean reads, and these were aligned to the maize reference genome (Zea mays B73_V4, ftp://ftp.ensemblgenomes.org/pub/plants/release-46/fasta/zea_mays/dna/) using HISAT2 (http://ccb.jhu.edu/software/hisat2, v. 2.0.1).” in Lines 493-496 of the manuscript. We then have re-submitted the manuscript.

Thank you for your consideration.

  1. DESeq1 or Seq2?

Thanks for your positive comments. In this study, we’re taking an R package called DESeq2 to analyze differentially expressed genes (DEGs). So, as suggested, the corresponding contents were revised that “DEGs were analyzed using an R Package called DESeq2 in Bioconductor (http://www.bioconductor.org/) for each comparison with the parameters of false discovery rate (FDR) < 0.05 and |log2 fold-change| >1.” in Lines 500-502 of the manuscript. We then have re-submitted the manuscript.

Thank you for your consideration.

  1. The abstract was not well prepared and needs to be rewritten.

Thanks for your positive comments. As suggested, we have re-written the abstract of the manuscript, namely “Lignin is an important factor affecting agricultural traits. The mechanism of lignin metabolism in maize (Zea mays) mesocotyl elongation was investigated during seed germination. Maize seed was treated with 4.16×10-3 M 24-epibrassinolide (EBR) and 3.05×10-3 μM brassinazole stimulation under 3 and 20 cm deep-seeding stress. Mesocotyl transcriptome sequencing together with targeted metabolomics analysis and physiological measurements were employed in two contrasting genotypes. Our results revealed differentially expressed genes (DEGs) were significantly enriched in phenylpropanoid biosynthesis, plant hormone signal transduction, flavonoid biosynthesis, and alpha-linolenic acid metabolism. There were 153 DEGs for lignin biosynthesis pathway, 70 DEGs for peroxisome pathway, and 325 differentially expressed transcription factors (TFs) of MYB, NAC, WRKY, and LIM were identified in all comparisons, and highly interconnected network maps were found among multiple TFs (MYB and WRKY) and DEGs for lignin biosynthesis and peroxisome. This caused p-coumaraldehyde, p-coumaryl alcohol, sinapaldehyde down-accumulation, and caffeyl aldehyde and caffeyl alcohol up-accumulation. The sum/ratios of H-, S-, and G-lignin monomers was also altered, which decreased total lignin formation and accumulation, resulting in cell wall rigidity decreased. As a result, a significant elongation of maize mesocotyl was detected under deep-seeding stress and EBR signaling. These findings provide information on the molecular mechanisms controlling maize seedling emergence under deep-seeding stress and will aid in the breeding of deep-seeding maize cultivars.” (Lines 11-28 ). We then have re-submitted the manuscript.

Thank you for your consideration.

Reviewer 3 Report

Title:

Transcriptomic and Metabolic Profiling Reveals a Lignin Metabolism Network Involved in Mesocotyl Elongation during Maize Seed Germination.

Major points:

  1. Fig 1 in the introduction is not required, instructions related to Fig 1 are also not required.
  2. In the results section, Fig 4,5,7,8 should be placed in the attached Fig. The results should be concise, not exhaustive.
  3. Fig 9 should be in the results section.

     4. Table 1 should be placed in the attached table.

Author Response

Dear Ms. Fiona Yan and Reviewers

Thank you for your letter of – and for the referee’s comments concerning our manuscript, “Transcriptomic and Metabolic Profiling Reveals a Lignin Metabolism Network Involved in Mesocotyl Elongation during Maize Seed Germination (plants-1646395)”. We have carefully studied these comments and have made corresponding corrections to the manuscript, which we describe in detail below. We would like to re-submit the manuscript and that for possible publication on the Special Issue: “Molecular Mechanism of Seed Germination under Different Environment Conditions” of Plants. Thank you very much for your time and consideration.

Your manuscript has now been reviewed by experts in the field. Please find your manuscript with the referee reports at this link: https://susy.mdpi.com/user/manuscripts/resubmit/ 716146146e9aabffde737744ead434c0. Please revise the manuscript according to the referees' comments and upload, the revised file within 10 days. Please use the version of your manuscript found at the above link for your revisions.

Thanks for the positive comments of you and all reviewers for our manuscript. As suggested, we have carefully revised the manuscript. We then have re-submitted the manuscript.

Thank you for your consideration.

(I) Any revisions to the manuscript should be marked up using the “Track Changes” function if you are using MS Word/LaTeX, such that any changes can be easily viewed by the editors and reviewers.

Thanks for the positive comments of you and all reviewers for our manuscript. As suggested, we have carefully revised the manuscript, and all corresponding contents were modified by “Track Changes” function of MS Word. We then have re-submitted the manuscript.

Thank you for your consideration.

(II) Please provide a cover letter to explain, point by point, the details of the revisions to the manuscript and your responses to the referees’ comments.

Thanks for the positive comments of you and all reviewers for our manuscript. As suggested, we have carefully revised the manuscript. In addition, we have prepared a detailed response letter to all reviewers for each point, and then have re-submitted the manuscript.

Thank you for your consideration.

(III) If you found it impossible to address certain comments in the review reports, please include an explanation in your rebuttal.

Thanks for the positive comments of you and all reviewers for our manuscript. As suggested, we have carefully revised the manuscript and have answered all questions of each reviewer point by point. We then have re-submitted the manuscript.

Thank you for your consideration.

(IV) The revised version will be sent to the editors and reviewers.

Thanks for the positive comments of you and all reviewers for our manuscript. As suggested, we have re-submitted the revised version of the manuscript.

 Thank you for your consideration.

Reviewer 3

Major points:

  1. Fig 1 in the introduction is not required, instructions related to Fig 1 are also not required.

Thanks for your positive comments. As suggested, the Fig. 1 has be placed in the file of supplement materials as Figure S1. We then have re-submitted the manuscript and supplement materials.

Thank you for your consideration.

  1. In the results section, Fig 4,5,7,8 should be placed in the attached Fig. The results should be concise, not exhaustive.

Thanks for your positive comments. As suggested, the Fig. 4, 5, 7, and 8 have be placed in the file of supplement materials as Figure S3, S4, S5, and S6, respectively. In addition, we have further simplified the results of the manuscript. We then have re-submitted the manuscript and supplement materials.

Thank you for your consideration.

  1. Fig 9 should be in the results section.

Thanks for your positive comments. As suggested, the Fig. 9 has be placed in the results section of the manuscript as Fig. 4, and the corresponding description about Fig. 4 was that “2.9. Quantitative real-time (qRT-PCR) analysis of DEGs. To further analyze the differences in DEGs at the transcriptional level, we verified the reliability of our RNA-Seq data using qRT-PCR and analyzed the relative expression level of 30 selected DEGs, including 10 DEGs in lignin biosynthesis pathway, 12 in TFs, and 8 DEGs in the peroxisome pathway (Figure 4A-4B).” in Lines 259-263 of the manuscript. We then have re-submitted the manuscript and supplement materials.

Thank you for your consideration.

  1. Table 1 should be placed in the attached table.

Thanks for your positive comments. As suggested, the Table 1 has be placed in the file of supplement materials as Table S2, respectively. We then have re-submitted the manuscript and supplement materials.

Thank you for your consideration.

In addition, Charlesworth Author Services (https://www.cwauthors.com.cn/) for providing linguistic assistance during the preparation of this manuscript, and other any modifications were performed by “Track Changes” function of MS Word.

Thank you for your consideration.

Sincerely,

Xiaoqiang Zhao professor

State Key Laboratory of Aridland Crop Science, Gansu Agricultural University

E-mail: zhaoxq3324@163.com

Round 2

Reviewer 1 Report

Major changes were introduced. Now article is better and easier to understanging. However, article should also contain conclusion section. Please add it.

Author Response

Dear Dr. Nopparat Suthprasertporn and Reviewers

Thank you for your letter of – and for the referee’s comments concerning our manuscript, “Transcriptomic and Metabolic Profiling Reveals a Lignin Metabolism Network Involved in Mesocotyl Elongation during Maize Seed Germination (plants-1646395)”. We have carefully studied these comments and have made corresponding corrections to the manuscript, which we describe in detail below. We would like to re-submit the manuscript and that for possible publication on the Special Issue: “Molecular Mechanism of Seed Germination under Different Environment Conditions” of Plants. Thank you very much for your time and consideration.

Your manuscript has now been reviewed by experts in the field. Please find your manuscript with the referee reports at this link: https://susy.mdpi.com/user/manuscripts/resubmit/ 716146146e9aabffde737744ead434c0. Please revise the manuscript according to the referees' comments and upload, the revised file within 4 days. Please use the version of your manuscript found at the above link for your revisions.

Thanks for the positive comments for our manuscript. As suggested, we have further revised and improved the manuscript. We then have re-submitted the manuscript.

Thank you for your consideration.

(I) Any revisions to the manuscript should be marked up using the “Track Changes” function if you are using MS Word/LaTeX, such that any changes can be easily viewed by the editors and reviewers.

Thanks for the positive comments for our manuscript. As suggested, we have further revised and improved the manuscript, and any modifications were marked by “Track Changes” function of MS Word. We then have re-submitted the manuscript.

Thank you for your consideration.

(II) Please provide a cover letter to explain, point by point, the details of the revisions to the manuscript and your responses to the referees’ comments.

Thanks for the positive comments for our manuscript. As suggested, we have further revised and improved the manuscript. In addition, we have prepared a detailed explanation for each questions, and have re-submitted the manuscript and response letter.

Thank you for your consideration.

(III) If you found it impossible to address certain comments in the review reports, please include an explanation in your rebuttal.

Thanks for the positive comments for our manuscript. As suggested, we have further revised and improved the manuscript. In addition, we have prepared a detailed explanation for each questions, and have re-submitted the manuscript and response letter.

Thank you for your consideration.

 (IV) The revised version will be sent to the editors and reviewers.

Thanks for the positive comments for our manuscript. As suggested, we have re-submitted the revised version of the manuscript.

 Thank you for your consideration.

If one of the referees has suggested that your manuscript should undergo extensive English revisions, please address this issue during revision. We propose that you use one of the editing services listed at https://www.mdpi.com/authors/english or have your manuscript checked by a native English-speaking colleague. Do not hesitate to contact us if you have any questions regarding the revision of your manuscript. We look forward to hearing from you soon.

Thanks for the positive comments for our manuscript. The English language has been well modified by Charlesworth Author Services (https://www.cwauthors.com.cn/). According to these suggestion of Charlesworth Author Services and all reviewers, we have tried our best to improve the English language of the manuscript. We then have re-submitted the manuscript.

Thank you for your consideration.

Reviewer 1

Major changes were introduced. Now article is better and easier to understanging. However, article should also contain conclusion section. Please add it.

Thanks for your affirmation of the previous modification of the manuscript. As suggested, we have re-added the Conclusion section of the manuscript, namely “5. Conclusions. The molecular mechanism of maize deep-seeding tolerance is very complex. By analyzing the transcriptomes, targeted metabolomics data, and physiological results of two maize genotypes (N192 and Ji853) MES, some biochemical pathways, in particular phenylpropanoid biosynthesis, and corresponding candidate genes were closely associated with maize deep-seeding tolerance. Further analysis detected 153 DEGs controlling lignin biosynthesis, 70 DEGs involving in peroxisome biogenesis, and 325 TFs (MYB, NAC, WRKY, and LIM) in four comparisons, and highly interconnected regulatory networks among multiple DEGs/TFs of MYB, WRKY, 4CL, C4H, F5H, CAD, CAT, SOD, ACSL, HAO, ACOX, ACAA1, PEX13, and PEX14 were also generated, which synergistically regulated the significant changes of intermediate metabolites accumulation (p-coumaraldehyde, p-coumaryl alcohol, sinapaldehyde, caffeyl aldehyde, and caffeyl alcohol), enzymes activity (SOD, POD, and CAT), and sum/ratios of H-, S-, and G-lignin monomers, and resulting in a significant MES elongation in maize under deep-seeding stress and EBR stimulation. This study provides a theoretical basis for further clarification of the complex regulation mechanism of deep-seeding tolerance of maize and the breeding of deep-seeding tolerant maize cultivars.” (Lines 548-564).” in Lines 540-556 of the manuscript. We then have re-submitted the manuscript.

Thank you for your consideration.

Reviewer 3

Moderate English changes required.

Thanks for your positive comments for our manuscript. The English language has been well modified by Charlesworth Author Services (https://www.cwauthors.com.cn/). According to these suggestion of Charlesworth Author Services, we have tried our best to improve the English language of the manuscript. We then have re-submitted the manuscript.

Thank you for your consideration.

Does the introduction provide sufficient background and include all relevant references?

Thanks for your positive comments for our manuscript. In the Introduction section of the manuscript, we analyzed in detail the biological functions of enzymes/genes involved in lignin biosynthesis from phenylpropanoid pathway, lignin function including hydrophobic properties, mineral transport, and responses to plant pathogens and environmental stresses, and other factors involved in lignin formation and metabolism (transcription factors, reactive oxygen species formation and scavenging, and peroxisomes). In addition, we also discussed the relationships among mesocotyl elongation, deep-seeding tolerance, and drought resistance in maize (Lines 32-93). These contents have laid a foundation for the implementation of our study. As suggested, we have further revised and improved the Introduction section of the manuscript (Lines 32-93). We then have re-submitted the manuscript.

Thank you for your consideration.

Is the research design appropriate?

Thanks for your positive comments for our manuscript. In this study, maize mesocotyls of deep-seeding tolerant N192 and intolerant Ji853 was incubated for 10 days at 3 cm sowing depth (CK), 20 cm sowing depth (DSS), 4.16×10-3 M 24-epibrassinolide [EBR; an active brassinosteroid (BR)] induction at 20 cm sowing depth (DSS+EBR), and 4.16×10-3 M EBR and 3.05×10-3 μM brassinazole (BRZ; an inhibitor of BR biosynthesis) induction at 20 cm sowing depth (DSS+EBR+BRZ) conditions to detect the differentially expressed genes (DEGs) involved in lignin biosynthesis pathway, peroxisomes biogenesis pathway, and differentially expressed transcription factors (TFs; MYB, NAC, LIM, WRKY), analyze the differences of intermediate metabolites and enzymes activities of superoxide dismutase (SOD), peroxidase (POD), and catalase (CAT) by RNA-sequencing, ultra-high-performance liquid chromatography-tandem mass spectrometry (UPLC-MS/MS), and physiological measurements. Therefore, The transcriptome and targeted metabolomics data generated here may help guide further research to develop novel strategies of promoting deep-seeding tolerance in maize from lignin metabolism. In the context of these facts, we have further revised and improved the materials and methods section of the manuscript (Lines 461-547). We then have re-submitted the manuscript.

Thank you for your consideration.

Are the methods adequately described?

Thanks for your positive comments for our manuscript. As suggested, we have further revised and improved the materials and methods section of the manuscript (Lines 461-547). We then have re-submitted the manuscript.

Thank you for your consideration.

Are the results clearly presented?

Thanks for your positive comments for our manuscript. As suggested, we have further revised and improved the results section of the manuscript (Lines 941-275). We then have re-submitted the manuscript.

Thank you for your consideration.

Are the conclusions supported by the results?

Thanks for your positive comments for our manuscript. We have re-added the conclusion section of the manuscript, namely “5. Conclusions. The molecular mechanism of maize deep-seeding tolerance is very complex. By analyzing the transcriptomes, targeted metabolomics data, and physiological results of two maize genotypes (N192 and Ji853) MES, some biochemical pathways, in particular phenylpropanoid biosynthesis, and corresponding candidate genes were closely associated with maize deep-seeding tolerance. Further analysis detected 153 DEGs controlling lignin biosynthesis, 70 DEGs involving in peroxisome biogenesis, and 325 TFs (MYB, NAC, WRKY, and LIM) in four comparisons, and highly interconnected regulatory networks among multiple DEGs/TFs of MYB, WRKY, 4CL, C4H, F5H, CAD, CAT, SOD, ACSL, HAO, ACOX, ACAA1, PEX13, and PEX14 were also generated, which synergistically regulated the significant changes of intermediate metabolites accumulation (p-coumaraldehyde, p-coumaryl alcohol, sinapaldehyde, caffeyl aldehyde, and caffeyl alcohol), enzymes activity (SOD, POD, and CAT), and sum/ratios of H-, S-, and G-lignin monomers, and resulting in a significant MES elongation in maize under deep-seeding stress and EBR stimulation. This study provides a theoretical basis for further clarification of the complex regulation mechanism of deep-seeding tolerance of maize and the breeding of deep-seeding tolerant maize cultivars.” (Lines 540-556). In order to this conclusion in this manuscript, we have described the results section (Lines 94-275) in detail and carried out in-depth the discussion section (Lines 548-564) based on previous studies. We then have re-submitted the manuscript.

Thank you for your consideration.

Sincerely,

Xiaoqiang Zhao professor

State Key Laboratory of Aridland Crop Science, Gansu Agricultural University

E-mail: zhaoxq3324@163.com

Reviewer 3 Report

No

Author Response

(The authors gave the same response as above.)
